# Tracking B Cell Memory to SARS-CoV-2 Using Rare Cell Analysis System

**DOI:** 10.3390/vaccines11040735

**Published:** 2023-03-26

**Authors:** Dong-Yan Tsai, Chun-Hung Wang, Perry G. Schiro, Nathan Chen, Ju-Yu Tseng

**Affiliations:** 1MiCareo Taiwan Co., Ltd., 5F, No. 69, Ln. 77, Xing Ai Rd., Neihu Dist., Taipei City 114, Taiwan; 2Adimmune Corporation, No. 3, Sec.1, Tanxing Rd., Tanzi Dist., Taichung City 427, Taiwan

**Keywords:** SARS-CoV-2, vaccines, rare cells, memory B cells

## Abstract

Rapid mutations within SARS-CoV-2 are driving immune escape, highlighting the need for in-depth and routine analysis of memory B cells (MBCs) to complement the important but limited information from neutralizing antibody (nAb) studies. In this study, we collected plasma samples and peripheral blood mononuclear cells (PBMCs) from 35 subjects and studied the nAb titers and the number of antigen-specific memory B cells at designated time points before and after vaccination. We developed an assay to use the MiSelect R II System with a single-use microfluidic chip to directly detect the number of spike-receptor-binding domain (RBD)-specific MBCs in PBMCs. Our results show that the number of spike-RBD-specific MBCs detected by the MiSelect R II System is highly correlated with the level of nAbs secreted by stimulated PBMCs, even 6 months after vaccination when nAbs were generally not present in plasma. We also found antigen-specific cells recognizing Omicron spike-RBD were present in PBMCs from booster vaccination of subjects, but with a high variability in the number of B cells. The MiSelect R II System provided a direct, automated, and quantitative method to isolate and analyze subsets of rare cells for tracking cellular immunity in the context of a rapidly mutating virus.

## 1. Introduction

Neutralizing antibody studies have been essential for both the development of vaccines for SARS-CoV-2 as well as in population-wide immunity surveillance. However, the continued emergence of variants with greater escape from antibody-mediated neutralization requires additional research into the cellular immune response to reduce mortality and prevalence of post-COVID-19 syndromes. Improved vaccine protection against SARS-CoV-2 will likely require advancements such as mosaic vaccines, nasal vaccines, or vaccines targeting broader binding, and a more detailed understanding of the cellular response to these approaches [1]. Enzyme-linked immunosorbent assays (ELISA), enzyme-linked immunospot (ELISpot) assays, intracellular cytokine staining, activation-induced marker assays, magnetic and fluorescence flow cytometry, and sequencing have provided extraordinary information on the cellular response to vaccination and infection [2,3]. Each of these technologies has its strengths and limitations and the complexity of many of the workflows limits some applications [4]. For example, one extraordinarily important study examining the effect of the pre-exposure T cell repertoire to the immune response post-vaccination cited that it was limited “due to the high experimental effort required for antigen-specific T cell enrichment and clonal tracking” and “to technical limitations in tracking low numbers of cells with low clonality” [5]. Memory B cells (MBCs) persist in circulation after vaccination or infection [6], but studies have suggested the waning in concentration of MBCs may be correlated with increased risk of SARS-CoV-2 infection [7]. In an attempt to develop a more reproducible and automated technology for the study of rare cells in circulation, we demonstrate the feasibility of using the MiSelect R II System to directly isolate and analyze spike-receptor-binding domain (RBD)-specific MBCs in peripheral blood mononuclear cells (PBMCs) at a series of time points from before vaccination to 6 months after a third shot of SARS-CoV-2 vaccine.

## 2. Methods

### 2.1. Vaccinated Subjects

Between 17 July and 27 November 2021, we recruited 35 SARS-CoV-2 naïve subjects in Taiwan who subsequently received two doses of SARS-CoV-2 vaccinations. No subjects reported having been previously infected with SARS-CoV-2, and because this study was conducted in Taiwan before a significant SARS-CoV-2 outbreak it is strongly believed that all subjects were SARS-CoV-2 naïve. Additionally, no binding antibodies (bAbs) or nAbs were detected in any of the subjects prior to vaccination (T0) The subjects had a median age of 35 and a range of 19 to 57 years. The 24 male and 11 female subjects were all believed to be immunocompetent, in general good health, and provided written informed consent. During this study all subjects received two doses of AZD-1222, BNT162b2, and/or mRNA-1273 SARS-CoV-2 vaccines. Most subjects received two doses of the same vaccine (AZD1222 and AZD1222, BNT162b2 and BNT162b2, mRNA-1273 and mRNA-1273) with the exception of 3 subjects who received AZD1222 for their first dose and mRNA-1273 for their second dose. Subject cohort information is described in Appendix A. In accordance with the vaccination policy of the Ministry of Health and Welfare in Taiwan, the second dose of the vaccine was administered approximately three months after the first dose prior to August 2021. However, after August 2021, the interval between the two doses was shortened to one month for the remaining 11 subjects. Human plasma samples and PBMCs were collected for serology analysis and cellular memory analysis at four time points. T0 is defined as being shortly before vaccination, T1 is 4 weeks after the first vaccine dose, T2 is 4 weeks after the second dose, and T3 is 8 weeks after the second dose. Most subjects also had samples collected at T4, which was 4 weeks after a booster dose, and T5, 6 months after a booster dose. 

### 2.2. Peripheral Blood Sample Processing

Venous blood was collected from vaccinated people into K_2_EDTA anti-coagulant tubes and centrifugated at 1600× *g* for 10 min to separate the plasma. The plasma was stored at −80 °C until it was needed for the SARS-CoV-2 binding and neutralizing antibody ELISA analyses. The plasma-reduced blood samples were diluted with ISOTON II buffer and transferred into 15 mL Ficoll-Paque (GE Healthcare Life Sciences, Chicago, IL, USA, 17144003). After standard density gradient centrifugation at 400× *g* for 30 min without use of the centrifuge break, the PBMC fraction was transferred into a new 50 mL tube, diluted with ISOTON II buffer, and recentrifuged. Aliquots containing 1 × 10^7^ PBMCs were cryopreserved in fresh 90% fetal bovine serum (FBS; Gibco, Waltham, MA, USA, A31605-01) and 10% dimethyl sulfoxide (DMSO; Sigma, St. Louis, MI, USA, D4540-500ML) at −80 °C.

### 2.3. Detecting SARS-CoV-2 Binding and Neutralizing Antibody 

An ELISA buffer kit (TONBO biosciences, San Diego, CA, USA, 41-9245-Kit) was used to test the plasma for the level of SARS-CoV-2 binding and neutralizing antibodies. Briefly, ELISA 96-well plates were coated with 0.1 μg per well SARS-CoV-2 spike-RBD recombinant protein (R&D system, Minneapolis, MN, USA, 10500-CV-100) or Omicron spike-RBD recombinant protein (R&D system, 11056-CV-100) in phosphate-buffered saline (PBS) at 4 °C overnight. The next day, plates were washed with a wash buffer containing 0.05% TWEEN-20 in PBS followed by a one hour incubation with a blocking buffer containing 1% BSA in PBS, and four rounds of washing with the wash buffer. Then, 100 μL of diluted plasma sample (at a 1:100 dilution for binding and neutralizing antibodies test) was added to each well and incubated for 1 h at room temperature. After plasma incubation, plates were washed once with wash buffer. For detecting SARS-CoV-2 binding antibody in plasma samples, 100 μL of 1:20,000 diluted HRP donkey anti-human IgG (BioLegend, San Diego, CA, USA, 410902) was added to each well and incubated for 1 h. For detecting SARS-CoV-2 neutralizing antibody in plasma samples, 50 μg of HRP conjugated (HRP conjugation Kit, BIO-RAD LNK006P) hACE2 recombinant protein (Abnova, Taipei City, Taiwan, P6639) in 100 μL PBS (100 μg spike-RBD with 50 μg hACE2; ~8:1 molar ratio) was added to each well and incubated for 1 h. Plates were again washed 7 times with wash buffer and developed with 100 μL of the 1-Step Ultra 3,3′,5,5′-Tetramethylbenzidine (TMB) substrate for 3 min at room temperature. The reaction was stopped with 100 μL of 1 M HCl stop solution. Plates were read with a Multiskan FC reader (Thermo Fisher, Waltham, MA, USA, 51119000) at 450 nm and 620 nm wavelengths. Monoclonal antibody CR3022 (abcam, ab273073) and WHO International Standard 20/136 for anti-SARS-CoV-2 immunoglobulin (NICBS) [8] were included on each plate to calculate the standard curve of relative antibody concentrations. CR3022 has been reported to bind with the RBD domain of SARS-CoV-2 and the binding epitope does not overlap with the ACE2 binding site [9]. The binding antibody assay used CR3022 in a range from 0.01 to 0.0001 μg/mL and 20/136 in a range from 0.625 to 2.5 BAU/mL. The neutralizing antibody assay used 20/136 in a range from 10 to 40 IU/mL. The assay was validated using the WHO standards 20/136, 20/142, 20/148, and 20/150. The WHO standard 20/136 has a known level of 1000 units and was diluted to create a standard curve. The binding antibody titer levels are listed in Appendix A, and the neutralizing antibody titer levels are in Appendix A. The WHO 20/150 standard has a geometric mean high level of anti-RBD IgG antibody titers per mL, with a reported 95% confidence interval (CI) of 663 to 1008 binding antibody units (BAU) and a reported neutralizing antibodies 95% CI of 1198 to 1912 international units (IU) per mL. The WHO 20/148 standard contains a middle level of anti-RBD IgG antibody titers, reported as 131 to 311 BAU/mL 95% CI, and 161 to 274 nAb IU/mL 95% CI. The WHO 20/142 standard is a negative plasma control with no detectable titers. By using the WHO 20/136 as a standard curve in our ELISA, the mean anti-RBD and nAb IgG titers of WHO 20/150 are 885.8 ± 100.2 BAU/mL and 1497.4 ± 290.4 IU/mL. The mean anti-RBD IgG and nAb titers of WHO 20/148 are 214.0 ± 24.7 BAU/mL and 245.7 ± 60.2 IU/mL.

### 2.4. Detecting SARS-CoV-2 Spike-RBD Specific Memory B Cells

Cryopreserved PBMCs were harvested and washed with ISOTON II buffer once at a density of 1 × 10^6^ cells/mL. The washed cells were resuspended in 100 µL ISOTON II buffer and incubated with DyLight 550 (DyLight 550 conjugation kit, Abcam, Taipei City, Taiwan, AB201800-3X10UG) conjugated to either wildtype spike-RBD or Omicron spike-RBD for 30 min at room temperature. After incubation, the PBMC samples were washed with ISOTON II buffer three times to remove the unbound antibody. Next, PBMC samples were loaded on the MiSelect R II system (MiCareo, Taipei City, Taiwan) to detect SARS-CoV-2 spike-RBD-specific MBCs. As described previously [10], the MiSelect R II system accepts up to 16 mL of minimally processed fluid such as whole blood or concentrated PBMCs and actively isolates targets expressing a specific biomarker. In this case, cells bound to SARS-CoV-2 spike-RBD recombinant proteins were isolated inside of the microfluidic SelectChip (MiCareo) using laser-induced fluorescence. The MiSelect R II System proceeds to automatically incubate the isolated cells with a chosen panel of fluorescence bound antibodies and image the cells using LED-induced fluorescence. The cell images are then analyzed with MiCyte cell analysis software (MiCareo), allowing for consistent memory B cell enumeration using the criteria of being positive for CD19-PerCP (clone 4G7), CD27-FITC (clone M-T271), IgG (Jackson ImmunoResearch, West Grove, PA, USA, 309-001-008), and SARS-CoV-2 spike-RBD or Omicron spike-RBD, and negative for CD3-APC (clone OKT3). The MiSelect R II system additionally allows for the output of target cells into a collection vial for subsequent analysis such as PCR and sequencing, but that mode of operation was not used in this study. 

### 2.5. In Vitro Differentiation of SRAS-CoV-2 Spike-RBD-Specific Memory B Cells to Antibody Secreting Cells

Vaccine-induced antigen-specific cells from PBMC samples are known to differentiate into antibody secreting cells [11,12,13]. Cryopreserved PBMCs were cultured at a density of 1 × 10^6^ cells/mL and stimulated with R848 (1 μg/mL; MabTech, Human IgG ELISpot kit, 3850-2A), IL-2 (10 ng/mL; MabTech, Nacka Strand, Sweden, Human IgG ELISpot kit, 3850-2A), IL-21 (100 ng/mL; PeproTech, 200-21), sCD40 ligand (1 μg/mL; PeproTech, 310-02), and SARS-CoV-2 spike-RBD (1 μg/mL; R&D system) and then maintained in RPMI 1640 (Gibco; Thermo Fisher Scientific, Waltham, MA, USA, 61870036) containing 10% FBS, and 1% penicillin/streptomycin (Gibco; Thermo Fisher Scientific, 10378016) for 3 days. Cultured supernatants were collected at day 3 and diluted at a 1:80 dilution. The diluted supernatant was analyzed with the same neutralizing antibody ELISA as was used for the plasma samples. 

## 3. Results

### 3.1. Antibody Responses of Vaccinated Subjects

We detected IgG nAb (Figure 1B–E) in plasma samples using our ELISA. Four weeks after the first vaccine dose (time point T1), but before the second dose, the level of nAb was increased both in subjects vaccinated with AZD-1222 and in subjects vaccinated with mRNA vaccines. Peak levels of nAb IgG were detected 4 weeks after the second vaccine dose (T2) (Figure 1B). Interestingly, 8 weeks after the second vaccine dose (T3), the nAb levels remained high but the mean had decreased, although not by a statistically significant amount (Figure 1E). At T5, 6 months after a booster shot, nAbs were mostly not detectable (Figure 5B). These results are consistent with other studies [14,15,16]. Two subjects vaccinated with AZD1222 did not have detectable levels of nAb at the T3 timepoint. Both subjects were at the older end of our study population (aged 55 to 57) (Figure 1B). Our ELISA assay to detect anti-RBD IgG in plasma (Appendix A–D) matched well with the nAb data shown in Figure 1. When measured as both BAU/mL and μg/mL of anti-RBD, the level correlated strongly with neutralizing titers against RBD-ACE2 binding (Appendix A–G). Together, these data demonstrated that the neutralizing antibody level could be detected by our ELISA and that it is highly correlated with the anti-RBD antibody level. These results also concur with many previous SARS-CoV-2 vaccine studies [14,15,16]. 

### 3.2. Spike-RBD-Specific Memory B Cells Detected by the MiSelect R II System

The MiSelect R II System was designed to find absolute numbers of low abundance cells such as SARS-CoV-2 antigen-specific cells, by requiring minimal sample preparation and utilizing microfluidic automation. We developed an assay using SARS-CoV-2 spike-RBD recombinant proteins conjugated with DyLight550 as probes to track spike-RBD specific MBCs in 10^6^ PBMC samples at different time points (Figure 2A). The MiSelect R II isolates SARS-CoV-2 spike-RBD positive cells in the microfluidic chip, followed by automatic incubation with a panel of fluorescently tagged antibodies and imaging at multiple wavelengths. Spike-RBD-specific MBCs were classified as being CD19+, CD27+, IgG+, spike-RBD+, and CD3- (Figure 2B and Appendix A). 

Before vaccination (time point T0), the baseline number of cells found by the MiSelect R II in all subjects was 20 ± 11 cells per 10^6^ PBMCs. However, we detected 91, 92, and 96 cells/10^6^ PBMCs in three individual SARS-CoV-2 naïve subjects before vaccination (Figure 2F). A recent study demonstrated that pre-existing immunity to seasonal endemic coronavirus could have profound consequences for memory B cell response to SARS-CoV-2 [17]. Other studies have identified large heterogeneity in T cell response based on pre-exposure T cell characteristics [2,18]. The pre-existing spike-RBD-specific MBCs found in these three subjects may be involved in cross-reactive immunity. Interestingly, the numbers of spike-RBD-specific MBCs in these three subjects increased significantly over time, potentially indicating that vaccination built cumulatively on their pre-existing cellular immunity. These three subjects also had PBMCs which secreted neutralizing antibodies prior to vaccination (Figure 3D). The heterogeneity of immune history is becoming larger with many people being infected multiple times with different variants. Tracking the cellular level of subtypes of B memory cells may provide insight into immunity on an individual level.

The observed number of spike-RBD MBCs increased dramatically in all subjects after prime and second dose vaccination (Figure 2C–F). The number of cells was found to be increased, but not by a significant amount, between time point T2 (4 weeks after the second dose) and T3 (8 weeks after the second dose (Figure 2F)). The number of cells further increased after a booster shot (T4) and remained high for the following 6 months (Figure 5C). Our results for the number and trend of spike-RBD MBCs agree with recent previous studies. Both virus infection and vaccination could induce plasmablasts and MBC populations, as measured by flow cytometry, that respond to secondary immune responses [19,20]. Recent studies indicated that CD19Low, CD20-, CD38+, spike-RBD+ specific plasmablasts could be rapidly induced 1 to 2 weeks after vaccination with BNT162b2 or mRNA-1273 [21,22]. However, the proportion of spike-RBD-specific plasmablasts will gradually decrease over time. On the other hand, CD19+, CD20+, CD27+, spike-RBD-specific MBCs were stable in peripheral blood months after mRNA vaccination [22]. 

### 3.3. Cellular Memory Responses from PBMCs of Vaccinated Subjects

To validate an in vitro culture assay to differentiate spike-RBD-specific cells into antibody secreting cells, we used PBMCs from both pre-vaccination subjects and convalescent subjects (Appendix A). Ranges of different cell densities were stimulated with R848, IL-2, IL-21, sCD40, and spike-RBD protein as described [12,13]. After 3 days of stimulation, the culture supernatants were collected for the neutralizing antibody ELISA analysis (Appendix A). Using a cell density of 10^5^ PBMCs per well yielded the highest nAb titers in the stimulated culture supernatants. In addition, the differentiation of spike-RBD antigen-specific plasma cells can be seen under a microscope (Appendix A). PBMCs were isolated from blood from vaccinated subjects at four timepoints (T0, T1, T2, and T3) and cultured with R848, IL-2, IL-21, sCD40, and spike-RBD protein. Supernatants were collected to measure the change in nAb levels over time (Figure 3A–C). As expected, high levels of secreting neutralizing antibody were observed 4 weeks after the first vaccine dose (T1). At T2, 4 weeks after the second vaccine shot, the level of secreted nAb was slightly, but not significantly, increased compared to T1. At 8 weeks after the second dose (T3), an increase was observed compared to T1 (Figure 3D). At T4 and T5, the B cells were still able to produce a high level of secreted nAb when stimulated (Figure 5D). Studies show both plasmablasts and MBCs secrete IgA, IgM, and IgG antibodies after vaccination [21,22]. However, for time points longer than 1 month post-vaccination, the antibody production is mainly contributed by MBCs. Taken together, these results demonstrate that the ability to secrete antibodies persists up to six months after a dose of vaccination. Spike-RBD-specific MBCs were induced by SARS-CoV-2 vaccines and rapidly secrete antibodies when they encounter matching foreign virus antigens for all but two of the 35 subjects.

### 3.4. Correlation between Antibody Response and Cellular Memory from Vaccinated Subjects

We investigated the potential correlation between the antibody response in plasma and the B cellular memory from PBMCs (Figure 4). First, we used data from 35 vaccinated subjects at time points T0 to T3 to calculate the relationship between nAb titers in plasma and nAb titers from stimulated PBMCs (Figure 4A). The Spearman’s rank correlation coefficient, *ρ*, was 0.62, which shows clear correlation, but is not particularly strong. At T5, 6 months after a booster shot, the nAb in plasma was not detectable, while the nAb from stimulated PBMCs continued to be high indicating no correlation (Figure 5B). The correlation between nAb in plasma and spike-RBD-specific MBCs from PBMCs was also similarly low, *ρ* = 0.69 (Figure 4B). Our results agree with a previous study that showed antibody levels did not strongly correlate with memory B cell frequencies in vaccinated SARS-CoV-2 naïve and vaccinated SARS-CoV-2 recovered individuals [23]. Next, we investigated the correlation between nAb from stimulated PBMCs versus the number of spike-RBD-specific MBCs over time. The *ρ* value of 0.79 demonstrates a strong correlation between the number of spike-RBD positive MBCs found by the MiSelect R II System and the titer of stimulated bulk PBMCs (Figure 4C). The correlation continued to be strong, *ρ* of 0.78 (data not shown), even at 6 months after a booster dose (Figure 5C). This finding helps validate the utility of the MiSelect R II assay and suggests the warranting of further investigation of subtypes of these rare cells.

### 3.5. Cellular Memory to Major Variants of Concern (VOC)

As SARS-CoV-2 continues to mutate, it is vital to monitor the elicited cellular immune response to the spike-RBD of VOC strains [24,25,26]. We applied the MiSelect R II System to investigating if PBMCs from vaccinated but not previously infected subjects would bind to Omicron spike RBD protein. PBMCs were collected 4 weeks (time point T4) after the third vaccine dose (Figure 6A). Three subjects who had received a first and second dose of AZD122 and three subjects who had received a first and second dose of mRNA-1273 were randomly chosen. All six subjects received mRNA vaccines for their third dose. We used wildtype spike-RBD or Omicron (B.1.1.529) spike-RBD protein conjugated with DyLight550 to detect CD19+, CD27+, IgG+, wildtype or Omicron spike-RBD+, CD3- cells with the MiSelect R II System. While antigen-specific cells that recognized the Omicron spike-RBD were detected in PBMCs from subjects with both homologous and heterologous vaccinations, the number of cells found in several of the subjects was quite low (Figure 6B). The low number of cells may indicate that the subject would have a weak immune response to the Omicron VOC and be at higher risk for severe disease. The ELISA for the nAb titer levels found in plasma showed an even greater loss of response to Omicron compared to wildtype (Figure 6C). Using the MiSelect R II System to detect and analyze the number of antigen-specific cells may be a direct and reproducible method to determine the level of immune cell protection against future mutations of the SARS-CoV-2 virus.

## 4. Discussion

Studies have shown SARS-CoV-2 vaccines induced robust anti-RBD and nAb IgG response to the spike-RBD of the virus [15,27,28,29,30]. As the SARS-CoV-2 virus continues to mutate to evade prior immunity obtained from both prior infections and vaccinations, the role of the cellular response becomes more critical in the understanding of both individual risk of disease severity and societal burden. Most studies of spike-RBD-specific cells have been conducted using magnetic or fluorescence flow cytometry [17,29,31]. Using flow cytometry to achieve a reproducible count of rare cells is challenging due to variable losses during sample preparation, uncertainty in gating cell types at the tail end of distributions, and Poissonian statistics dictating the need to analyze a large volume of samples [4,32,33,34].

The MiSelect R II System used a mostly automated workflow to successfully track the number of cells positive for multiple antibodies from 35 subjects from before vaccination to six months post-vaccination. These data suggest that using the MiSelect R II System could detect CD19+, CD27+, IgG+, spike-RBD+, CD3-, spike-RBD-specific memory B cell numbers directly. These antigen-specific MBCs could be induced by prime dose SARS-CoV-2 vaccine (time point T1) and increased stably with second dose vaccination (time point T2 and T3) and a booster shot (T4) and can remain high for the following 6 months. While nAb was mostly not detectable in plasma 6 months after a booster shot, the MiSelect R II continued to find binding immune cells. Additionally, the number of cells present correlated with the level of nAb secreted from the bulk stimulated PBMCs.

Tracking the cellular response to vaccination is an important consideration for monitoring immune escape by variants. Previous studies have shown that MBC numbers remain relatively stable for up to 9 months, but that a reduction in concentration may lead to greater susceptibility to infection [6,7]. It is not yet known how the number of these cells will correlate to protection against severe disease from SARS-CoV-2 BA.2, BA.4, or BA.5 subvariants, such as XBB.1.5, but moving to more sophisticated correlates of protection that go beyond just nAb measurements is becoming increasingly urgent [4]. The MiSelect R II is relatively simple and reduces the technology burden allowing this assay to be more readily applied to larger studies to monitor cellular response during the development of new vaccines, correlation with disease severity, or efficacy of therapies.

### Limitations of the Study

Our study relies on a small study population which is not representative of Taiwan or the rest of the world. The study population consisted of a narrow range of ages and included only generally healthy people. Some studies have demonstrated a negative association between sex and vaccination-induced antibody response after SARS-CoV-2 mRNA vaccination [25,27,35,36]. We analyzed gender differences in plasma nAb concentrations, stimulated PBMC secreting nAb, and spike-RBD-specific MBCs before the second vaccine dose (time point T1), and after the second dose vaccination (time point T2) (Appendix A). We found that there was no significant difference in nAb in plasma, stimulated PBMCs, or spike-RBD-specific MBCs from PBMC by sex.

## Figures and Tables

**Figure 1 vaccines-11-00735-f001:**
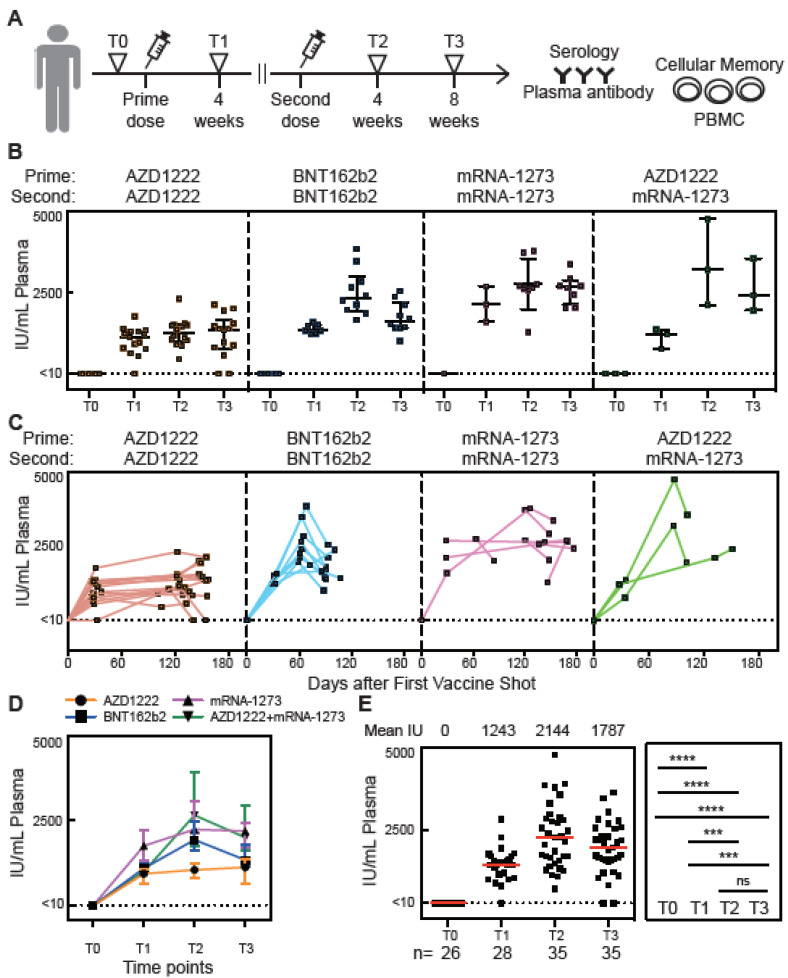
Neutralizing antibody responses before and after vaccination in SARS-CoV-2 naïve subjects. (**A**) Timeline for blood draws for analysis of nAbs in plasma, the secretion from stimulated PBMCs, or cellular identification with the MiSelect R II. T0 is defined as being shortly before vaccination, T1 is 4 weeks after the first vaccine dose, T2 is 4 weeks after the second dose, and T3 is 8 weeks after the second dose. The time interval between the first dose and the second dose varied from approximately three months to just one month. (**B**–**D**) Concentration of neutralizing anti-spike-RBD IgG antibodies in plasma samples from vaccinated subjects over time. (**E**) Total neutralizing anti-spike-RBD IgG concentrations and mean number in plasma sample from all vaccinated subjects (n) for each time point. IU is international units. The error bar indicates the median and interquartile range. Statistics were calculated using the non-parametric Mann–Whitney test. The dotted line indicated the limit of detection (LOD) for the assay.; *** *p* < 0.001; **** *p* < 0.0001; ns, no significant difference.

**Figure 2 vaccines-11-00735-f002:**
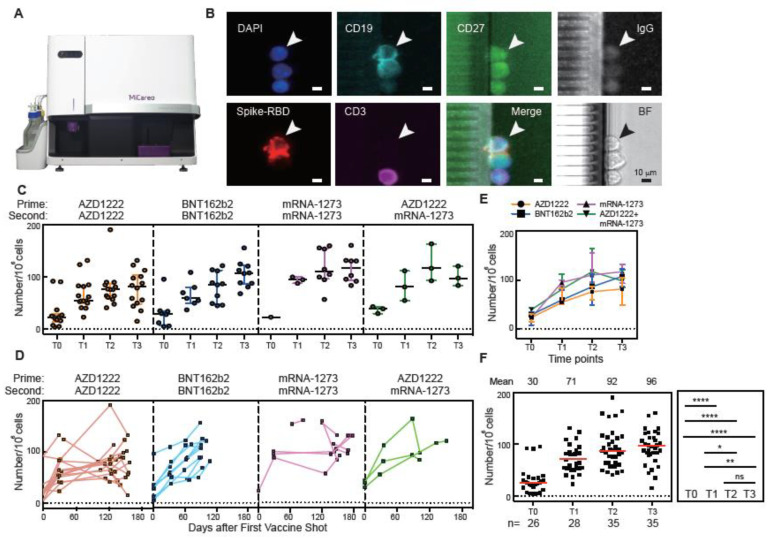
Spike-RBD-specific memory B cells induced by SARS-CoV-2 vaccination over time. (**A**) The MiSelect R II System for spike-RBD cell isolation and analysis. (**B**) Example images of the automated reagent labeling and fluorescence imaging performed by the MiSelect R II. (**C**–**E**) CD19+, CD27+, IgG+, spike-RBD+, CD3- specific MBCs in PBMC samples from vaccinated subjects over time. (**F**) Total number of spike-RBD-specific MBCs in PBMC samples from all vaccinated subjects for each time point. The number of subjects (n) was shown below the graph. The error bar indicates the median and interquartile range. Statistics were calculated using the non-parametric Mann–Whitney test. The dotted line indicated the limit of detection (LOD) for the assay. * *p* < 0.05; ** *p* < 0.01; **** *p* < 0.0001; ns, no significant difference.

**Figure 3 vaccines-11-00735-f003:**
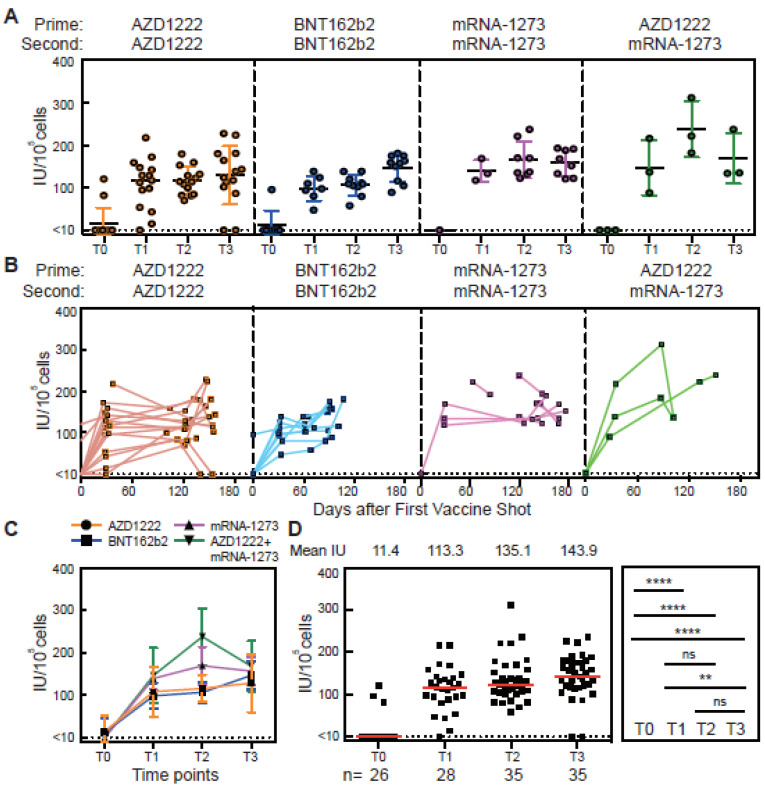
In vitro culture system to differentiate spike-RBD-specific cells into antibody secreting cells. (**A**–**C**) Concentration of neutralizing anti-spike-RBD IgG antibodies in culture supernatant over time from PBMC stimulated with R848, IL-2, IL-21, sCD40, and WT-RBD. (**D**) Total neutralizing anti-spike-RBD IgG concentrations in culture supernatant after 3 days of stimulation for each time point. The number of subjects (n) is shown below the graph. IU indicates the international units. The error bar indicates that the median and interquartile range. Statistics were calculated using the non-parametric Mann–Whitney test. The dotted line indicated the limit of detection (LOD) for the assay. ** *p* < 0.01; **** *p* < 0.0001; ns, no significant difference.

**Figure 4 vaccines-11-00735-f004:**
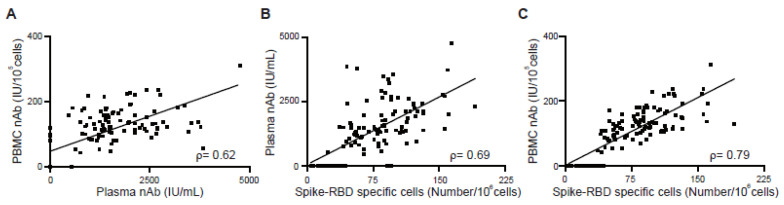
Correlation between neutralizing antibody titers in plasma or secreting by PBMC stimulation and spike-RBD-specific memory B cells. (**A**) Correlation of neutralizing IgG in plasma with neutralizing IgG levels from in vitro stimulation. (**B**) Correlation of neutralizing IgG in plasma with spike-RBD-specific MBCs in PBMC samples. (**C**) Correlation of neutralizing IgG levels from in vitro stimulation with spike-RBD-specific MBCs in PBMC samples. Correlations were calculated using non-parametric Spearman’s rank correlation and are shown with linear trend lines.

**Figure 5 vaccines-11-00735-f005:**
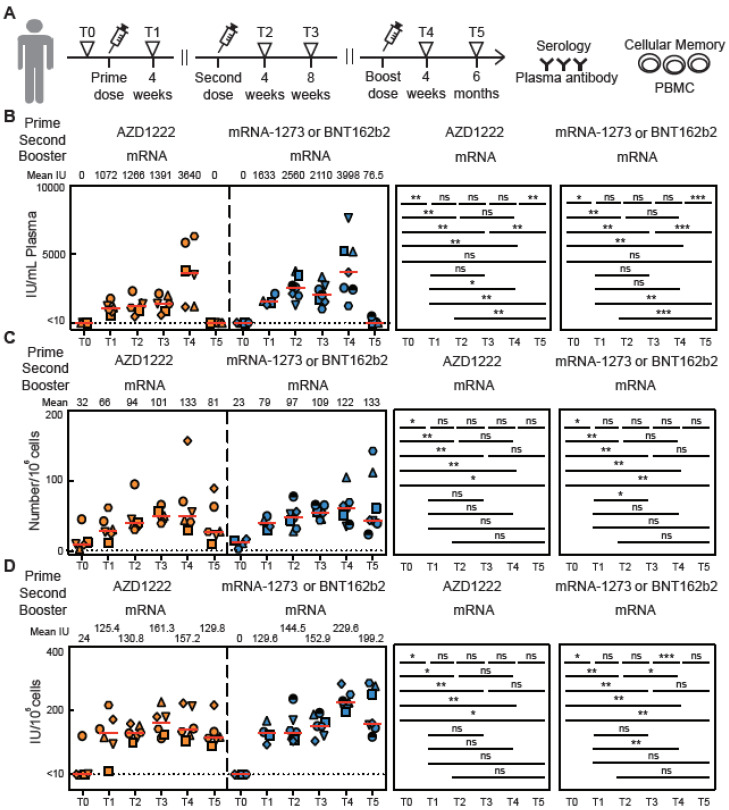
Spike-RBD-specific cells induced by SARS-CoV-2 vaccination over time. (**A**) Timeline for blood draws for analysis of nAbs in plasma, the secretion from stimulated PBMCs, or cellular identification with the MiSelect R II. T4 was 4 weeks after a 3rd vaccination, while T5 was a blood draw 6 months later. (**B**) Concentration of neutralizing anti-spike-RBD IgG antibodies in plasma samples from vaccinated subjects over time. (**C**) Total number of spike-RBD-specific MBCs in PBMC samples from all vaccinated subjects for each time point. (**D**) Concentration of neutralizing anti-spike-RBD IgG antibodies in culture supernatant over time from PBMC stimulated with R848, IL-2, IL-21, sCD40, and WT-RBD. The error bar indicates the median. Statistics were calculated using the non-parametric Mann–Whitney test. The dotted line indicated the limit of detection (LOD) for the assay. * *p* < 0.05; ** *p* < 0.01; *** *p* < 0.001; ns, no significant difference.

**Figure 6 vaccines-11-00735-f006:**
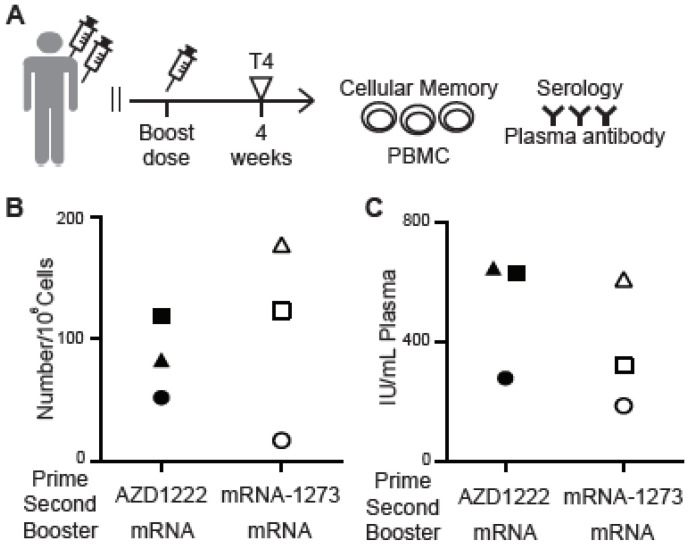
Omicron spike-RBD-specific memory B cells detected by the MiSelect R II system after boost vaccination. (**A**) Blood was collected 4 weeks after a 3rd vaccine shot (T4). (**B**) CD19+, CD27+, IgG+, wildtype or Omicron spike-RBD+, CD3- specific MBCs in PBMC samples from mRNA boost vaccination (n = 6). (**C**) Concentration of neutralizing anti-wildtype-RBD or anti-Omicron-RBD IgG antibodies in plasma sample from mRNA boost vaccination (n = 6).

## Data Availability

The data presented in this study are available on request from the corresponding author.

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
