# Peer review of "Tracking B Cell Memory to SARS-CoV-2 Using Rare Cell Analysis System"

_vaccines, 2023, doi:10.3390/vaccines11040735_

Round 1

Reviewer 1 Report

Vaccines 2249851

Title should indicate B cell memory. Likewise, abstract mentions T and B cell memory but only B cells were assayed; thus, they need to be modified appropriately. The use of the MiSelect R II to “isolate” seems somewhat misleading; it can separate them for further analysis within the systems’s instrumentation, but it is unclear if after being separated they can be collected for further evaluation outside of the instrument.

Line 26-27: this reviewer completely agrees with the need for additional need for research into the cellular responses, but this needs to be explained why.

line 44-45: “conducted in Taiwan before significant SARS-CoV-2” ; this is not an accurate statement, there are reports of COViD-19 occurring in Taiwan before July 17 2021 (Gong et al. Emerging Microbes Infections 9:1, 1457-1466, DOI: 10.1080/22221751.2020.1782271there is also CDC reports of COVID-19 in Taiwan before July 2021). The study may be flawed, because it seems that it was assumed the people enrolled were SARS-CoV-2 naïve; there is no indication those enrolled were assayed for virus or Ab prior to vaccination. 

Detecting anti-SARS-CoV-2 and neutralizing antibody assay is not described well including use of donkey ant-human HRP, which is incorrect and catalog number should be given. Likewise, the neutralizing antibody assay is inadequately described.  Antibody dilutions should be described. Source and catalog numbers should be provided for all reagents used.

Line 89: Monoclonal antibody CR3022 also is inadequately described Abcam lists multiple catalog products as CR3022

Line 98: all abbreviations should be spelled out first time, e.g., nAb 

Line 124: describe R848

Line 132: wrong word order – we detected IgG nAb

Line 134: Peak levels of nAb IgG were detected 4 weeks after the second vaccine dose (T2) is confusing because figure (Fig.1C) shows second doses of different vaccines but timely is based on days after first vaccine shot and BNT162b2 peaked at 60 days after first shot and AZD1222 followed by mRNA-1273 at about 90 days, the other two vaacines had too much variance for any conclusion about peak timing. Figure 1A should also clarify different times between first and second vaccinations in the legend or on Fig. 1B&C.

Line 136: “remained high but had decreased a not significant amount”  should be -remained high at T2 and did not significantly decrease by T3 (Fig.1E). 

Line 136-137: “At T3, 6 months after a booster shot, nAb were mostly not detectable (Fig. 5B)” This is wrong – but not clear if its time (T) or something else. 

All statements made in text need to be supported by indicating where data is found, e.g. “Two subjects vaccinated with AZD1222 did not have detectable 138 levels of nAb at T3 timepoint “

Line 295: the funding issue needs to be corrected, If the reseachers were funded by MiCareo then the research was funded by MiCareo.

If the manuscript was written in a clearer fashion, it might be more useful although the major point seems to be more a promotion of the technology (the MiCareo system) than the novelty of the results. Without documentation of screening for presence of the virus in the 35 participants or for the presence of any antibodies to Spike-RBD of SARS-CoV-2 prior to any vaccination some of the conclusions being made cannot be accepted since it is known that the virus was in Taiwan prior to July 17 2021. The major novelty of the analysis is as the title would suggest – the memory B cell evaluation, there is already much known about antibody titers after infection and/or vaccinations. Thus, the technology would have been more useful for further evaluation of the memory B cells, e.g., differences in proportions of B cells with different epitope specificities or even if there were any different proportions of isotypes or affinity differences with the different SARS-CoV-2 BA.2, BA.4, or BA.5 subvariants. It was good to see that there was good correlation between plasma nAb levels and numbers of memory B cells but many of the conclusions have been reported for prior nAb analyses; the authors should have taken more advantage if their instrumentation for the cellular analysis. 

Author Response

We thank the reviewer for the very detailed and helpful comments. Please the attachment for our revisions.

Manuscript ID: vaccines-2249851

Tracking Cellular Memory to SARS-CoV-2 Using Rare Cell Analysis System

Dong-Yan Tsai , Chun-Hung Wang , Perry Gershon Schiro , Nathan Chen , Ju-Yu Tseng

We thank the reviewer for their comments and have revised the manuscript to address the suggestions in nearly all cases.

Reviewer 1

Title should indicate B cell memory.

Response: We have changed the title to indicate it is specific to B cells: “Tracking B Cell Memory to SARS-CoV-2 Using Rare Cell Analysis System”

Likewise, abstract mentions T and B cell memory but only B cells were assayed; thus, they need to be modified appropriately.

Response: The first sentence of the abstract mentions the need to study both T and B cells, and as correctly pointed out this study only looked at B cells. While it is certainly important to also study T cells, we have removed the reference to the T cells from this sentence to avoid confusion as suggested.

The use of the MiSelect R II to “isolate” seems somewhat misleading; it can separate them for further analysis within the systems’s instrumentation, but it is unclear if after being separated they can be collected for further evaluation outside of the instrument.

Response: The MiSelect R II can be used in several different modes. The mode of operation used in this study confines the target cells inside of the microfluidic chip where they can be incubated with reagents and imaged by the instrument at different wavelengths. A different mode of operation, that was not used in this study, outputs the target cells into a vial, well plate, or other container for additional analysis outside of the instrument such as PCR or sequencing. For both modes the cells are “isolated” from the bulk population of cells, but the reviewer is correct to point out that “isolated” means very different things depending on the way the system is operated. We have added a sentence in the methods section “Detecting SARS-CoV-2 spike-RBD specific memory B cells”, to note the possibility to isolate target cells into an external collection vial.

Line 26-27: this reviewer completely agrees with the need for additional need for research into the cellular responses, but this needs to be explained why.

Response: We have added a sentence and a reference in the introduction sections to highlight the importance of further research into the cellular response particularly because of the effects of mutations on immune evasion to vaccination and the resulting continued mortality and risk of long covid.

line 44-45: “conducted in Taiwan before significant SARS-CoV-2” ; this is not an accurate statement, there are reports of COViD-19 occurring in Taiwan before July 17 2021 (Gong et al. Emerging Microbes Infections 9:1, 1457-1466, DOI: 10.1080/22221751.2020.1782271; there is also CDC reports of COVID-19 in Taiwan before July 2021). The study may be flawed, because it seems that it was assumed the people enrolled were SARS-CoV-2 naïve; there is no indication those enrolled were assayed for virus or Ab prior to vaccination. 

Response: We agree that SARS-CoV-2 was present in Taiwan prior to the samples collected in this study but stand by our statement that “it is strongly believed that all subjects were SARS-CoV-2 naïve” for the following reasons: (1.) Each subject was tested before vaccination (T0) and no neutralizing antibodies were found in any of the subjects (Fig 1B). (2.) None of the subjects had traveled outside of Taiwan during the pandemic. (3.) Taiwan had reported approximately 16,338 cumulative covid cases prior to our sample collection and 1,698 of those cases were imported rather than being domestic spread (CDC press release Oct 16 2021: https://www.cdc.gov.tw/En/Bulletin/Detail/fkgnStebmnhRl7jfW3kNVg?typeid=158). While confirmed cases are an undercount of true cases, the difference in Taiwan during this time period was substantially smaller than in most of the rest of the world due to aggressive contact tracing and testing policy. Even if undiagnostic cases raised the total to 30,000 cases it would still only represent about 0.1% of the population. We have added a sentence to clarify that the subjects were assayed before vaccination.  

Detecting anti-SARS-CoV-2 and neutralizing antibody assay is not described well including use of donkey ant-human HRP, which is incorrect and catalog number should be given.

Response: We have reworded parts of this section to clarify the assay protocol and added the catalog number for the diluted donkey anti-human HRP (BioLegend, 410902).  

Likewise, the neutralizing antibody assay is inadequately described.  Antibody dilutions should be described. Source and catalog numbers should be provided for all reagents used.

Response: We have added dilutions and catalog numbers for each reagent.

Line 89: Monoclonal antibody CR3022 also is inadequately described Abcam lists multiple catalog products as CR3022

Response: We have added information on CR3022 to indicate its purpose and listed the ab273073 catalog number.

Line 98: all abbreviations should be spelled out first time, e.g., nAb

 Response: We have modified the text to spell out abbreviations at their first usages when missing.

Line 124: describe R848

Response: R848 is an imidazoquinoline and agonist compound with potent anti-viral activity [Pubmed: 12032557]. It has been reported that R848 plus Interleukin (IL)-2 could stimulate vaccinated human memory B cell to product antigen-specific antibodies [Pubmed: 23454005] and is discussed in reference 12.

Line 132: wrong word order – we detected IgG nAb

Response: The wording order has been corrected as suggested.

Line 134: Peak levels of nAb IgG were detected 4 weeks after the second vaccine dose (T2) is confusing because figure (Fig.1C) shows second doses of different vaccines but timely is based on days after first vaccine shot and BNT162b2 peaked at 60 days after first shot and AZD1222 followed by mRNA-1273 at about 90 days, the other two vaacines had too much variance for any conclusion about peak timing. Figure 1A should also clarify different times between first and second vaccinations in the legend or on Fig. 1B&C.

Response: Time since a particular dose is a typical way to measure response, particularly for vaccinations since the interval between doses is not constant due either to policy, vaccine manufacturer, or personal choice. In the methods section we state: “In accordance with the vaccination policy of the Ministry of Health and Welfare in Taiwan the second dose of the vaccine was administers approximately three months after the first dose prior to August 2021. However, after August 2021, the interval between the two doses was shortened to one month for the remaining 11 subjects.” Figure 1A shows the specific time point definitions in relation to the dose and also a gap between the prime dose and the second dose. Figure 1B shows the response at specific time points and nearly all of the paper discusses the response in relation to this measurement method. Figure 1C is scientifically less helpful for looking at overall response because “days after first vaccine shot” is not a constant related to the timing of the second or booster shots. Figure 1C is included to highlight individual changes and show those changes per person over a time period measured in days. Figure 1C is included because there may be additional information to be learned from it, but the quantitative results and discussion in this paper are based on time points defined as time since a particular dose as shown in figure 1A. We have added info to note that we are discussing figure 1B in that paragraph and have added info to the figure legends to help clarify this timing.

Line 136: “remained high but had decreased a not significant amount”  should be -remained high at T2 and did not significantly decrease by T3 (Fig.1E). 

Response: We thank the reviewer for noticing this and have changed it to: “Interestingly, 8 weeks after the 2nd vaccine dose (T3) the nAb levels remained high but the mean had decreased, although not by a statistically significant amount (Fig. 1E)”

Line 136-137: “At T3, 6 months after a booster shot, nAb were mostly not detectable (Fig. 5B)” This is wrong – but not clear if its time (T) or something else. 

Response: What we wrote is certainly wrong! “T3” should have been “T5”, and this has now been corrected.

All statements made in text need to be supported by indicating where data is found, e.g. “Two subjects vaccinated with AZD1222 did not have detectable 138 levels of nAb at T3 timepoint “

Response: This sentence has been updated to indicate the data is shown in figure 1B.

Line 295: the funding issue needs to be corrected, If the reseachers were funded by MiCareo then the research was funded by MiCareo.

Response: By stating “This work was not funded by any outside sources.”, were meant to express that all work was funded and conducted by MiCareo as that is where the authors are employed and was not funded by a federal grant or other source. We have changed the funding to state “This work was funded by MiCareo.”

If the manuscript was written in a clearer fashion, it might be more useful although the major point seems to be more a promotion of the technology (the MiCareo system) than the novelty of the results. Without documentation of screening for presence of the virus in the 35 participants or for the presence of any antibodies to Spike-RBD of SARS-CoV-2 prior to any vaccination some of the conclusions being made cannot be accepted since it is known that the virus was in Taiwan prior to July 17 2021. The major novelty of the analysis is as the title would suggest – the memory B cell evaluation, there is already much known about antibody titers after infection and/or vaccinations. Thus, the technology would have been more useful for further evaluation of the memory B cells, e.g., differences in proportions of B cells with different epitope specificities or even if there were any different proportions of isotypes or affinity differences with the different SARS-CoV-2 BA.2, BA.4, or BA.5 subvariants. It was good to see that there was good correlation between plasma nAb levels and numbers of memory B cells but many of the conclusions have been reported for prior nAb analyses; the authors should have taken more advantage if their instrumentation for the cellular analysis. 

Response: We thank the reviewer for the very helpful and detailed comments. We have addressed the concern regarding subjects prior possible exposure to SARS-CoV-2 above. We are submitting this paper for publication exactly because we agree that further evaluation of memory B cells is important and that we believe this technology may be helpful in that research. We hope that this paper will bring awareness of this technical approach to researchers who have proposals such as those you list, so that they may more successfully conduct their research.

Reviewer 2 Report

This study collected plasma and PBMCs from 35 subjects before and after SARS-CoV-2 vaccination to study nAb titers and the number of antigen-specific memory B cells. The researchers directly detected the number of spike-RBD specific memory B cells in PBMCs using the MiSelect R II System. They found a strong correlation between the number of spike-RBD specific memory B cells and the level of nAbs secreted by stimulated PBMCs. This manuscript is well written. The results are consistent with many previously published studies. Although the researchers claim the MiSelect R II system is superior to the flow cytometry, this manuscript mainly utilized the MiSelect R II system to count the spike-RBD specific MBCs number in PBMC, which seems no different from what the normal flow cytometry could do.

1.     The researchers utilized the bulk PBMCs to test the cellular memory responses. Although Figure 4C showed the correlation of spike-RBD positive MBCs number with the titer of stimulated bulk PBMCs, it’s interesting to see the titer directly from the sorted MBCs culture. Could the MiSelect R II system sort the plasmablast cells or the memory B cells for the sub-culture? If yes, could the sub-culture be sorted as a single cell into a 96-well plate?

2.     Figure 6, it’s unclear the immune response difference between the wildtype versus the Omicron spike-RBD, although lines 236-237 the researchers claimed the subject had a weak immune response to the Omicron VOC. For the same subject, direct immune response comparisons are needed.

Author Response

We thank the reviewer for the important clarifying comments. Please the attachment for our revisions. 

Manuscript ID: vaccines-2249851

Tracking Cellular Memory to SARS-CoV-2 Using Rare Cell Analysis System

Dong-Yan Tsai , Chun-Hung Wang , Perry Gershon Schiro , Nathan Chen , Ju-Yu Tseng

We thank the reviewer for their comments and have revised the manuscript to address the suggestions in nearly all cases.

Reviewer 2

This study collected plasma and PBMCs from 35 subjects before and after SARS-CoV-2 vaccination to study nAb titers and the number of antigen-specific memory B cells. The researchers directly detected the number of spike-RBD specific memory B cells in PBMCs using the MiSelect R II System. They found a strong correlation between the number of spike-RBD specific memory B cells and the level of nAbs secreted by stimulated PBMCs. This manuscript is well written. The results are consistent with many previously published studies.

  1. Although the researchers claim the MiSelect R II system is superior to the flow cytometry, this manuscript mainly utilized the MiSelect R II system to count the spike-RBD specific MBCs number in PBMC, which seems no different from what the normal flow cytometry could do.

Response: Flow cytometry has been utilized quite successfully to count spike-RBD specific MBCs, but there are specific limitations such as the difficulty in obtaining a reproducible cell count number when the absolute number of cells is quite low. 

  1. The researchers utilized the bulk PBMCs to test the cellular memory responses. Although Figure 4C showed the correlation of spike-RBD positive MBCs number with the titer of stimulated bulk PBMCs, it’s interesting to see the titer directly from the sorted MBCs culture. Could the MiSelect R II system sort the plasmablast cells or the memory B cells for the sub-culture? If yes, could the sub-culture be sorted as a single cell into a 96-well plate?

Response: The MiSelect R II can sort cells for further culture, although some cells are cultured more successfully than others. The sorted cells are output pooled together into a single vial or well. To conduct single cell studies, individual cells can be picked out with a capillary, but that was not done in this study. We have added a sentence to note this in the paper.

  1. Figure 6, it’s unclear the immune response difference between the wildtype versus the Omicron spike-RBD, although lines 236-237 the researchers claimed the subject had a weak immune response to the Omicron VOC. For the same subject, direct immune response comparisons are needed.

Response: Clarifying the differences in immune response is very important and we thank the reviewer for noting this lack of direct comparison. Previously we had separated the data between Figure 5 and Figure 6 and direct comparison was not easily seen. We have now added data to Figure 6 to directly show the much lower nABs levels for Omicron vs wildtype. 

Reviewer 3 Report

The present manuscript describes “Tracking Cellular Memory to SARS-CoV-2 Using Rare Cell Analysis System”. SARS‑CoV‑2 belongs to the broad family of viruses known as coronaviruses.  It is a positive-sense single stranded RNA virus with a single linear RNA segment. Pfizer and Moderna had developed vaccines to effectively inhibit SARS‑CoV‑2. There have been millions of people affected and died by SARS‑CoV‑2 throughout the world. As the SARS-CoV-2 virus continues to mutate to evade prior immunity obtained from both prior infections and vaccinations, the role of the cellular response becomes more critical in the understanding of both individual risk of disease severity and societal burden. The authors have developed a new method to track cellular memory to SARS-CoV-2 using rare cell analysis system. The authors collected plasma samples and PBMCs from 35 subjects and studied the nAb titers and the number of antigen-specific memory B cells at designated time points before and after vaccination. Most of the studies for Spike-RBD specific cells have been done using magnetic or fluorescence flow cytometry. However, when the number of target cells is relatively low, it is difficult to get an exact cell count using flow cytometry and often fails to find reproducible number of cells. The authors developed an assay to use the MiSelect R II System with a single-use microfluidic chip to directly detect the number of spike-RBD specific memory B cells in PBMCs. The results showed that the number of spike-RBD specific memory B cells detected by the MiSelect R II System is highly correlated with the level of nAbs secreted by stimulated PBMCs, even 6 months after vaccination when nAbs were generally not present in plasma. They also found that the antigen-specific cells recognizing Omicron spike-RBD were present in PBMCs from booster vaccination of subjects, but with a high variability in the number of cells. The MiSelect R II System provided a direct, automated, and quantitative method to isolate and analyze subsets of rare cells for tracking cellular immunity in the context of a rapidly mutating virus. I would recommend to publish this excellent work in Vaccines Journal.

Author Response

We thank the reviewer for their kind words and review of our submission. We hope this work may provide additional tools for the important study of cellular response in the context of changing population immunity. 

Round 2

Reviewer 1 Report

Line 17: word order - (nAb) studies; 

Line 25 should probably be – number of B cells.

Line 56: Most readers of this journal would likely understand the nAb definition vs Ab, but for clarity the authors should make the assay clearer if they truly assayed nAb or only Ab to SARS-CoV-2. It is not made clear if binding Ab or only nAb was assayed prior to vaccination.  Since this is being used to attest to no prior exposure, it should be both.  Furthermore, this statement (Line 56) contradicts the statement later (Line 185 & 192) about no nAb before vaccination. 

Additionally, the methods do partially describe the assay for binding Ab and for nAb; however this also needs to be corrected. Line 95-96 used HRP-labeled donkey anti-human IgG not anti-human HRP. For nAb assay, (Line96-99) it needs to be made clear that the plasma to be tested for nAb was added (dilution?), incubated time, and washed (x times) before addition of  HRP-hACE2.

This report is clearly, in part, to promote the MiSelect R II System and in doing so the authors attempt to suggest (Line 275-277) that its better than using flow cytometry. This is inadequately supported, in that there are many reports indicating flow cytometry is very useful; it is the clinical methodology for assessing minimal residual disease in cancer patients who have very low numbers of cells being reproducibly assayed. There also are papers supporting its use for vaccination evaluation. The references used for making the authors point are lacking since they address other types of evaluations. 

Line 317 “may” should be removed. The study was supported mainly by their company 

Author Response

We thank the reviewer for highlighting areas to be clarified.

We thank the reviewer for their comments and have revised the manuscript.

Reviewer 1

Line 17: word order - (nAb) studies; 

Response: We corrected this word order.

Line 25 should probably be – number of B cells.

Response: We added “B” to clarify that this variability is in the B cells.

Line 56: Most readers of this journal would likely understand the nAb definition vs Ab, but for clarity the authors should make the assay clearer if they truly assayed nAb or only Ab to SARS-CoV-2. It is not made clear if binding Ab or only nAb was assayed prior to vaccination.  Since this is being used to attest to no prior exposure, it should be both.  Furthermore, this statement (Line 56) contradicts the statement later (Line 185 & 192) about no nAb before vaccination. 

Response: We have added text to clarify that both a binding assay and a neutralizing assay were done prior to vaccination and specified that the data is shown in figure 1B for the neutralizing antibody assay and in the supplementary Figure 1A for the binding antibodies. In Line 185 and 192 we are discussing finding cells (not antibodies) that did attach to the spike from samples taken prior to vaccination. The existence of cells that are able to bind is consistent with the other references listed in the next few lines showing that some subjects may have broadly responsive memory B cells due to prior exposures to other viruses. These three subjects all had no measurable binding or neutralizing antibodies prior to vaccination. When the PBMCs from these three subjects were stimulated they secreted neutralizing antibodies, which agrees with our finding that some cells bind to the spike. It appears that these memory B cells both recognize the spike and secrete antibodies to it when stimulated, but that is different than finding antibodies in the blood.

Additionally, the methods do partially describe the assay for binding Ab and for nAb; however this also needs to be corrected. Line 95-96 used HRP-labeled donkey anti-human IgG not anti-human HRP.

Response: We thank the reviewer for pointing out the potential confusion and have moved “HRP” to the beginning of the phrase as suggested. To clarify our intended meaning here is the description from the manufacturer (Biolegend): “HRP Donkey anti-human IgG (minimal x-reactivity) Antibody. This polyclonal donkey anti-human IgG antibody reacts with the heavy chains of human IgG and with the light (kappa and lambda) chains common to most human immunoglobulins. The polyclonal antibody was purified from donkey antiserum by human immunoglobulin affinity chromatography.”

For nAb assay, (Line96-99) it needs to be made clear that the plasma to be tested for nAb was added (dilution?), incubated time, and washed (x times) before addition of  HRP-hACE2.

Response: Lines 92-94 describe the dilution, incubation time, and wash times for the nAb assay: “100 mL of diluted plasma sample (at a 1:100 dilution for binding and neutralizing antibodies test) was added to each well and incubated for 1 hour at room temperature. After plasma incubation, plates were washed once with wash buffer.” 

This report is clearly, in part, to promote the MiSelect R II System and in doing so the authors attempt to suggest (Line 275-277) that its better than using flow cytometry. This is inadequately supported, in that there are many reports indicating flow cytometry is very useful; it is the clinical methodology for assessing minimal residual disease in cancer patients who have very low numbers of cells being reproducibly assayed. There also are papers supporting its use for vaccination evaluation. The references used for making the authors point are lacking since they address other types of evaluations.

Response: This work is certainly intended to demonstrate the feasibility of the MiSelect R II System at detecting certain cells with the hope that researchers will find value in both the system’s workflow and cellular lower count sensitivity. We are not claiming that the system is better than flow cytometry (FC) for most applications. Lines 275-277 as stated are specific to the use case where the number of cells available is below the commonly reproducible utility of flow cytometry. It is documented that FC struggles to accurately provide an absolute cell count at low numbers. One causal, non-peer reviewed, overview of the statistics with example applications can be found here: https://www.escca.eu/images/publication/PAR-08_Brando_ESCCA_Rare_Event_Analysis_2019_revised.pdf

We have reworded that sentence to make it more clear that the comment is limited to rare cell populations and added two references to the manuscript to better support the claim that FC statistically struggles as the cell prevalence falls (https://doi.org/10.3389/fimmu.2020.02169) and the specific use case of memory B cells (https://doi.org/10.3389/fimmu.2019.01271).

Minimal (or measurable) residual disease is an excellent example of the promise and limitations of flow cytometry. For example EuroFlow,  LeukemiaNet MRD Working Party, FNIH MRD (https://fnih.org/what-we-do/programs/biomarkers-consortium-measurable-residual-disease-acute-myeloid-leukemia-mrd) and other groups have been working for many years on trying to standardize and offer protocols for MRD in acute myeloid leukemia (AML). All of these groups have highlighted the extreme challenge of reproducibly identifying the target cells with enough sensitivity (https://doi.org/10.1182/blood.2021013626). For example: “Nonetheless, our previous data showed a low diagnostic performance in this method, mainly determined by low sensitivity, justifying the high rate of relapse associated to MRD negative patients (Rossi et al., 2012; Rossi et al., 2014). Some challenges affecting the detection of MRD by FC have been already addressed, while others still remain to be clarified.” (https://doi.org/10.1002/cyto.b.21855). 

Line 317 “may” should be removed. The study was supported mainly by their company 

Response: “May” has been removed as suggested.